# Diversity in *Coffea arabica* Cultivars in the Mountains of Gorongosa National Park, Mozambique, Regarding Bean and Leaf Nutrient Accumulation and Physical Fruit Traits

Niquisse J. Alberto [1], José C. Ramalho [2,3,*], Ana I. Ribeiro-Barros [2,3], Alexandre P. Viana [4], Cesar A. Krohling [5], Sional S. Moiane [6], Zito Alberto [6], Weverton P. Rodrigues [7] and Fábio L. Partelli [8,*]

1   Breeding Program, Federal University of Espírito Santo, S/N Guararema, Alegre, Vitoria 29375-000, ES, Brazil
2   Plant Stress & Biodiversity Lab, Forest Research Center (CEF) and Associate Laboratory TERRA, Instituto Superior de Agronomia (ISA), Lisboa University (ULisboa), Quinta do Marquês, Av. República, 2784-505 Oeiras, Portugal
3   Geobiociences, Geoengeneering and Geotechnologies Unit (GeoBioTec), Faculty of Science and Technology (FCT), New University of Lisboa (UNL), Monte de Caparica, 2829-516 Caparica, Portugal
4   Laboratory of Plant Breeding, Center of Agricultural Science and Technology, Darcy Ribeiro State University of Northern Rio de Janeiro, Av. Alberto Lamego, 2000, Campos dos Goytacazes 28013-602, RJ, Brazil
5   Espírito Santo Research Institute, Technical Assistance and Rural Extension (Incaper), Vitória 29052-010, ES, Brazil
6   Gorongosa National Park, Goinha 2112, Mozambique
7   Center of Agricultural, Natural and Literary Sciences, State University of the Region of Tocantina do Maranhão (UEMASUL), Brejo do Pinto Ave., Estreito 65975-000, MA, Brazil
8   Department of Agrarian and Biological Sciences (DCAB), Center of Northern Espírito Santo (CEUNES), Federal University of Espírito Santo (UFES), Rod. BR 101 Norte, Km 60, Bairro Litorâneo, São Mateus 29932-540, ES, Brazil
*   Correspondence: cochichor@isa.ulisboa.pt (J.C.R.); partelli@yahoo.com.br (F.L.P.)

**Abstract:** Genetic characteristics and their interaction with environmental conditions, including nutritional management, determine coffee productivity and quality. The objective of this study was to evaluate fruit traits and nutrient accumulation in the fruit, husk, and bean, as well as in the leaves of different *Coffea arabica* cultivars cropped in the Gorongosa National Park, Mozambique. The experiment evaluated nine coffee cultivars in a randomized block design, with four replicates. Fruit and leaf samples were collected over two months (June and July 2021), in the fruit maturation phase, oven-dried and analyzed, namely, through a clustering unweighted pair group method with arithmetic mean (UPGMA). The characterization of ripe and dried coffee bean indicated differences in the performance of the cultivars. The accumulation of the macronutrients N, K, and Ca and micronutrients Fe, Mn, and B was highest in the bean, husk, fruit, and leaves of the evaluated cultivars. Nutrient concentrations and accumulation in the different evaluated organs have a direct influence on the nutritional crop management. This is crucial for a nutritional diagnosis that ensures high yields, but such mineral levels are also a result of the existing genetic diversity among cultivars, which must be taken into account for management and breeding purposes.

**Keywords:** arabica coffee; characterization of fruit and bean; mineral nutrition; variability

## 1. Introduction

Coffee is socially and economically one of the most highly valued products in the world, representing an important source of revenue and employment in several tropical countries [1]. Coffee is native to Africa and belongs to the *Rubiaceae* family, with approximately 660 genera and, to date, 130 species [2]. Of these, *C. arabica* L. (Arabica coffee) and *C. canephora* Pierre ex A. Froehner (Conilon and Robusta coffee) dominate the international market. In 2021, ca. 10.2 million tons of coffee was harvested globally, of which ca. 59% was from *C. arabica* L. and 41% from *C. canephora* Pierre ex A. Froehner [1]. Brazil

and Vietnam are the largest coffee producers, but the African continent has more than 10 coffee-producing countries, with Ethiopia ranking first, followed by Uganda, Ivory Coast, Tanzania, Kenya, and Angola, with Mozambique on the way to becoming a promising new frontier in the coffee sector.

Mozambique is a large country with a high regional diversity of soil climatic conditions. The country produces coffee of the species *C. arabica*, *C. canephora* (Café Conilon), *C. zanguebariae*, and *C. racemosa* (locally known as Inhambane coffee), of which the latter two are considered native [3–6]. This makes necessary the development of high-yielding and well-adapted genotypes for each region, as well as demand studies on the genetic diversity of cultivars. Only then will it be possible to accurately recommend the use of region-specific, superior, and divergent cultivars with region-specific adaptation. In the mountainous region of the Gorongosa National Park (GNP), Sofala province, the cultivation of *C. arabica* was only recently initiated. Nevertheless, the activity has already become socially and economically relevant with regard to employment, resources, and foreign exchange income. Additionally, this coffee is cultivated in agroforestry system (AFS) management with native trees, contributing to rainforest reforestation, essential for the life quality of local communities [7].

While other coffee species are diploid, *Coffea arabica* is tetraploid and has four basic chromosome sets (n = 11), i.e., with a total of 44 chromosomes. This resulted from a recent evolutionary whole-genome duplication between *C. canephora* (Robusta) and *C. eugenioides* on the Central Ethiopia plateau [8,9]. The breeding system of *C. arabica* is classified as autogamous, with the wind and insect-mediated pollination of ovules occurring with pollen from the same plant, with an auto fecundation up to 90% of the flowers [10].

The physical and chemical attributes of fruit and bean, with potential implications for quality, were then examined. These included color, size, special flavor, phenolic content, soluble solids, chlorogenic, caffeic and *p*-coumaric acids, caffeine, trigonelline, lipids, and minerals. Most of these parameters were mainly affected by temperature (although without a strong negative impact on bean quality), and only marginally, if at all, by elevated [$CO_2$] [7,11–13].

The high cost of coffee cultivation under adequate soil and climate conditions is mainly due to crop management, primary processing, and pest and disease control [14]. Coffee farming has been modernized with significant technological advances in recent years, which have resulted in annual yield increases and cost savings, due to the monitoring and control of storage, organization, and agricultural activities [15]. Yet the ongoing climate changes and global warming are believed to already affect the current plantations with additional external constraints, mainly associated with supra-optimal temperatures and reduced water availability, mostly caused by reduced rainfall amounts and changes in their distribution pattern along the year [14]. To obtain segregating populations of *Coffea* species, the narrow genetic variation and low recombination capacity due to repeated self-fecundation processes represent major limitations [16–18]. On the other hand, due to the positive effect of elevated air $CO_2$ (usually associated with global warming) on plant C-metabolism, and the use of elite coffee cultivars that exhibit greater environmental resilience, the extent of the negative environmental impacts associated with climate changes might be less dramatic than previously estimated in the context of climate change and global warming [19,20].

The nutritional status of plants can be determined by the nutrient content in plant tissues, so that the physiologically active organs of bean, husk, and leaf are used for nutritional diagnosis. The correct interpretation of nutritional analysis is key to defining the adequate nutrient amounts to supply the crop, reduce input waste, improve the nutritional balance of the plants, and, consequently, to increase productivity [21–23]. The nutritional status of a crop is strongly influenced by the species and varieties, management practices, altitude, and maturity level, among others [24]. Additionally, mineral uptake by coffee trees also varies within a plant according to location, time of year, age, organs and tissues [25], and throughout the maturation cycle [25–27], and might be also impacted by environmental

conditions, namely, of high temperature [20]. Moreover, under conditions causing stomatal closure (water stress, low temperatures), the transpiration flow will be reduced and, consequently, nutrient translocation from the roots to the upper parts of the plant will be reduced as well [28]. However, the uptake efficiency of some nutrients is also genetically defined [29]. Several studies have pinpointed differences in the nutritional concentrations between *C. arabica* genotypes under the same management conditions [30,31]. This fact is due to the wide intra- and inter-specific extent of variability between *C. arabica* genotypes, especially in relation to characteristics such as growth, maturation cycle, nutrient accumulation, and stress tolerance [32]. Thus, this diversity can be exploited for the identification of genotypes better adapted to the range of edaphoclimatic conditions of the producing regions [30].

The objective of this study was to evaluate the fruit traits and nutrient accumulation in the bean, husk, fruit, and leaves of different *Coffea arabica* cultivars in the mountains of Gorongosa National Park, Mozambique.

## 2. Materials and Methods

### 2.1. Brief Characterization of Plant Material, Experimental Design, and Plant Area

The trial was conducted in the Gorongosa Mountains, in the Gorongosa National Park (GNP), one of the main conservation and reforestation areas of Mozambique, in the district Gorongosa, Sofala province, central Mozambique (longitude $34°04'05''$ E, latitudes $-18°47'99''$ to $-18°48'05''$ S; mean altitude 961 m asl). According to Köppen–Geiger, the mean annual temperature of the region is 29 °C, the mean annual rainfall is 1697 mm, and the climate is tropical savanna (Aw) and subtropical dry winter (Cw). The local soil was studied and the texture classified as clayey with wavy relief, based on the texture triangle [33]. The soil's chemical and physical properties are listed in Table 1.

**Table 1.** Chemical and granulometric characteristics of the soil of the experimental area of Gorongosa Mountain, Gorongosa National Park, Sofala, Mozambique.

| Chemical Properties | | | Results |
|---|---|---|---|
| M.O. | Organic matter (Oxy-Red.) | $dag/dm^3$ | 5.5 |
| Ph | (water—ratio 1:2.5) | unit | 5.1 |
| P | (Mehlich$^{-1}$) | $mg/dm^3$ | 8.4 |
| K | (Mehlich$^{-1}$) | $mg/dm^3$ | 80 |
| Ca | (Kcl$^{-1}$ mol/L) | $cmolc/dm^3$ | 1.2 |
| Mg | (Kcl$^{-1}$ mol/L) | $cmolc/dm^3$ | 0.4 |
| Al | (Kcl$^{-1}$ mol/L) | $cmolc/dm^3$ | 0.6 |
| H + Al | (Calcium acetate) | $cmolc/dm^3$ | 11 |
| S.B. | (Sum of bases) | $cmolc/dm^3$ | 1.81 |
| C.T.C. | (Cation exchange capacity at pH = 7) | $cmolc/dm^3$ | 12.81 |
| V% | (Base saturation) | % | 14 |
| %K C.T.C | (%K at CTC) | % | 2 |
| %Ca C.T.C | (%Ca at CTC) | % | 9 |
| %Mg C.T.C. | (%Mg at CTC) | % | 3 |
| %Al C.T.C. | (%Al at CTC) | % | 4.7 |
| %H+Al C.T.C. | (%H + Al at C.T.C.) | % | 86 |
| P (Resin) | | $mg/dm^3$ | 13.6 |
| P-rem | (Remaining phosphorus) | mg/L | 13 |
| Na | (Mehlich$^{-1}$) | $mg/dm^3$ | 3 |
| S | (Monocalcium acetic phosphate) | $mg/dm^3$ | 22 |
| B | (Hot water) | $mg/dm^3$ | 0.2 |
| Zn | (Mehlich$^{-1}$) | $mg/dm^3$ | 1.4 |
| Mn | (Mehlich$^{-1}$) | $mg/dm^3$ | 30.6 |
| Cu | (Mehlich$^{-1}$) | $mg/dm^3$ | 4.9 |
| Fe | (Mehlich$^{-1}$) | $mg/dm^3$ | 28 |

**Table 1.** *Cont.*

| Chemical Properties | Results |
|---|---|
| **Granulometric fractions (g kg$^{-1}$)** | |
| Sand | |
| Silt | |
| Clay | |
| Soil type | Clayish |

The nine *C. arabica* cultivars were sown in November 2018 (Table 2), arranged in a randomized complete block design with four replications, where the treatments corresponded to the different cultivars. The crop was grown in full sun at a spacing of 2 m between rows and 1.5 m between plants (2 × 1.5), with 3 m$^2$ per plant, equivalent to 3333 plants per hectare. The crop treatments consisted of hand weeding only, without the application of fertilizers, insecticides, or fungicides, in respect of the conservation status of the experimental area.

**Table 2.** Identification of the nine *Coffea arabica* L. cultivars, grown in the Gorongosa Mountain, Gorongosa National Park. Sofala, Mozambique.

| Identification | Name | MAPA Registration Number | Maturation Season | Productivity |
|---|---|---|---|---|
| 1 | Catuaí Vermelho IAC 44 | 2929 | Medium and late | High |
| 2 | Catucaí Amarelo 2SL | 4915 | Medium | High |
| 3 | Catuaí Amarelo IAC 39 | 2937 | Late | High |
| 4 | Catucaí 785-15 | 4996 | Precocious | High |
| 5 | Catucaí Vermelho 19/8 | 4909 | Precocious | High/Medium |
| 6 | Catuaí Vermelho IAC 81 | 2932 | Medium | High |
| 7 | Acauã | 4995 | Late | High |
| 8 | Catimor 128 | | | |
| 9 | Called Costa Rica | | | |

So far, the cultivars Catimor 128 and Costa Rica have not been registered by the Brazilian Ministry of Agriculture, Livestock and Food Supply (MAPA).

*2.2. Fruit and Leaf Collection*

When the plants were approximately 32 months old, samples were collected between June and July 2021. Per plot, 200 g fruit (relative fresh weight (RFW), i.e., 800 g per treatment/cultivar) was packed into paper bags. The material was dried in a convection oven at 65 °C. Subsequently, the fruits were then manually depulped and 100 g samples of bean per plot were prepared. All dry samples were evaluated at a moisture content of 11% (11%MC). The evaluations included the percentage of bean in the fruit (%B), husk percentage in the fruit (%S), dry weight per bean 11%MC (DBW), ripe fruit weight (RFW), dry fruit weight 11%MC (DFW), ratio of ripe fruit weight/dry fruit weight (RFW/DFW), ratio of ripe fruit weight/dry bean weight (RFW/DBW), ripe fruit weight/ton of bean 11%MC, ripe fruit weight/60 kg bag, and dry coffee weight/ton of 11%MC.

Leaf samples (15 leaves per plot) were collected, i.e., a total of 60 leaves per treatment (cultivar), taken between rows from either side of the plants, in the middle third of the plants exposed to the sun, consisting of the third or fourth pair of leaves from the apex of the plagiotropic branches. The material was placed in paper bags and dried to constant weight at 65 °C in a convection oven for approximately 72 h.

### 2.3. Mineral Concentration in Fruit and Leaf Tissues

The mineral concentration in the collected materials (fruit and leaf) was quantified separately for bean, husk, and leaf. The contents of nitrogen (N), phosphorus (P), potassium (K), calcium (Ca), magnesium (Mg), sulfur (S), iron (Fe), zinc (Zn), copper (Cu), manganese (Mn), and boron (B) were assessed according to the previously described methodology [34]. The N content was determined after sulfuric acid digestion using the Nessler colorimetric method; P by molecular absorption spectrophotometry; K by flame photometry; and S by sulfate turbidimetry. The contents of Ca, Mg, Fe, Zn, Mn, and Cu were determined by atomic absorption spectrophotometry, and B by colorimetry using the Azometin-H method. The nutrients P, K, Ca, Mg, S, B, Fe, Zn, Mn, and Cu were quantified by ICP-OES, after digestion with concentrated $HNO_3$ and $H_2O_2$ in an open digestion system. The ICP conditions were: plasma gas 8.0 L min$^{-1}$, auxiliary gas 0.70 L min$^{-1}$, and carrier gas 0.55 L min$^{-1}$ [35–37].

The dry weight of the husks and beans in a 60 kg bag of processed coffee was inferred from the ratio of bean to husk. The accumulation of macro- and micronutrients in the husks and beans was calculated as dry weight × concentration of each nutrient. The macro- and micronutrients accumulated in the fruit were calculated based on the sum of these nutrients contained in the husks and beans.

Moisture was determined by oven-drying four repetitions of each sample to a fully dry state (105 °C, for 24 h) [38]. Bean moisture is usually considered with 11% humidity, therefore a correction from 0 to 11% moisture (Mo) was needed, using the following equation:

$$\text{initial W.} * (100 - \text{initial Mo}) = \text{final W.} * (100 - \text{final Mo})$$

where initial W. = weight at 65 °C 170 h$^{-1}$, final W. = weight at 105 °C 24 h$^{-1}$, initial Mo = P. Initial—P. Final, and V Final Mo = 88.

Based on these calculations, the bean weight 11%MC (i.e., dry bean weight (DBW)) and the proportion of ripe fruit to dry bean weight 11%MC (RFW/DSW) were determined.

### 2.4. Statistical Analyses

The data were subjected to analysis of variance and the characteristics with significantly different results among cultivars, at a level of 5% probability ($p \leq 0.05$) by the F test, were compared by Tukey's HSD mean comparison test, at the same significance level. The data were analyzed using the software Genes [39].

The following statistical model was used:

$$\text{Yij} = \mu + \text{Gi} + \text{Bk} + \text{eijk}$$

where:

Yijk = observation regarding the i-th genotype in the j-th repetition;
$\mu$ = general constant;
Gi = effect of the i-th genotype, i = 1, 2, . . . , 9;
Bk = kth block effect, k = 1, 2, 3, 4;
eijk = experimental error.

The data matrix for cluster analysis consisted of 15 variables, namely, % of bean in the fruit, dry bean weight 11%MC, dry fruit weight 11%MC, kg of ripe fruit/60 kg of bean 11%MC, and total accumulation of N, P, K, Ca, Mg, S, Cu, Fe, Mn, Zn, and B in the fruit per ton of bean 11%MC.

To generate the dendrogram, the Mahalanobis generalized distance (D$^2$) (1936) was estimated by the following formula:

$$D^2_{ii'} = \sum_{j=1}^{n} . \sum_{j'=1}^{n} . W_{JJ'} d_j d_j$$

where:

$D^2_{ii'}$ is the Mahalanobis generalized distance between genotypes i and i', i = 1, 2, ... 9;
n = number of variables;
$W_{JJ'}$ = element of the j-th row and j-th column of the inverse variance matrix and residual covariances between the genotypes;
$d_j$ = difference between the means of the j-th variable in the two studied genotypes.

To establish a cut-off point in the dendrogram and define the number of groups, the Monjena procedure (1977) was used, based on the relative size of the merger levels (distances) in the dendrogram.

The UPGMA (unweighted pair group method with arithmetic mean) method was used for clustering to obtain the highest cophenetic correlation coefficient (0.84). The analysis was based on the following formula:

$$d_{(ij)k} = \text{mean}\left(d_{ik}; d_{jk}\right) = \frac{d_{ik} + d_{jk}}{2}$$

where $d_{(ij)k}$ is given by the average distances between pairs of individuals (i and k) and (j and k).

To visualize the dendrogram, we used the R software package "Circlize" [40].

The relative contribution of the characteristics was calculated from the method proposed by Singh [41], based on the partition of the total estimates of distances $D^2_{ii'}$, considering all possible pairs of individuals, with regard to each characteristic. The data were analyzed using R software [39].

## 3. Results

### 3.1. Dissimilarity between Genotypes and Genetic Contribution

The genetic diversity relative contribution was determined using the Singh method [41], resulting in values between 0.02% (fruit weight) and 44.45% (boron). The variables of bean weight (6.71%), micronutrient concentrations B (44.45%), Fe (20.58%), macronutrient S (10.48%), and macronutrient K (4.91%) were those that significantly contributed to the observed differences; that is, they were the variables that better expressed the existing genetic diversity, altogether representing 87.13% of the variability among the cultivars (Table 3).

**Table 3.** Relative contribution of bean, husk, leaves, fruit characteristics, and nutrient concentration and accumulation in nine *Coffea arabica* L. cultivars grown in the mountain region of the Gorongosa National Park, Sofala, Mozambique, according to Singh's method [41], based on the Mahalanobis generalized distance (D2).

| Variable | S.j | Value (%) | Cumulative Value (%) |
| --- | --- | --- | --- |
| B | 5559.12 | 44.45 | 44.45 |
| Fe | 2573.03 | 20.58 | 65.03 |
| S | 1310.84 | 10.48 | 75.51 |
| Bean weight | 839.46 | 6.71 | 82.22 |
| K | 614.21 | 4.91 | 87.13 |
| Mn | 482.87 | 3.86 | 91.00 |
| Mg | 278.87 | 2.23 | 93.23 |
| %Bean | 269.01 | 2.15 | 95.38 |
| ripe Kg | 214.37 | 1.71 | 97.09 |
| P | 145.69 | 1.17 | 98.26 |
| N | 101.21 | 0.81 | 99.07 |
| Ca | 70.37 | 0.56 | 99.63 |
| Cu | 23.63 | 0.19 | 99.82 |
| Zn | 20.87 | 0.17 | 99.98 |
| Fruit weight | 2.02 | 0.02 | 100.00 |

S.j: value proposed by Singh [41].

The dendrogram was constructed based on genotype grouping by the hierarchical cluster analysis UPGMA, using Mahalanobis' generalized distance as a dissimilarity measure, with regard to the variables % of bean in the fruit, bean weight 11%MC, dry fruit weight 11%MC, ripe fruit weight per 60 kg of bean 11%MC, and total nutrient accumulation (kg) in the fruit per ton of bean 11%MC. At the maximum cut-off limit of 250 of the dissimilarity between cultivars, the formation of two groups was observed (Figure 1). Group I included seven cultivars (Catucaí Amarelo 2SL, Catuaí Amarelo IAC 39, Catucaí 785-15, Catucaí Vermelho 19/8, Catuaí Vermelho IAC 81, Acauã and Catimor 128), and group II only included two cultivars (Costa Rica and Catuaí Vermelho IAC 44).

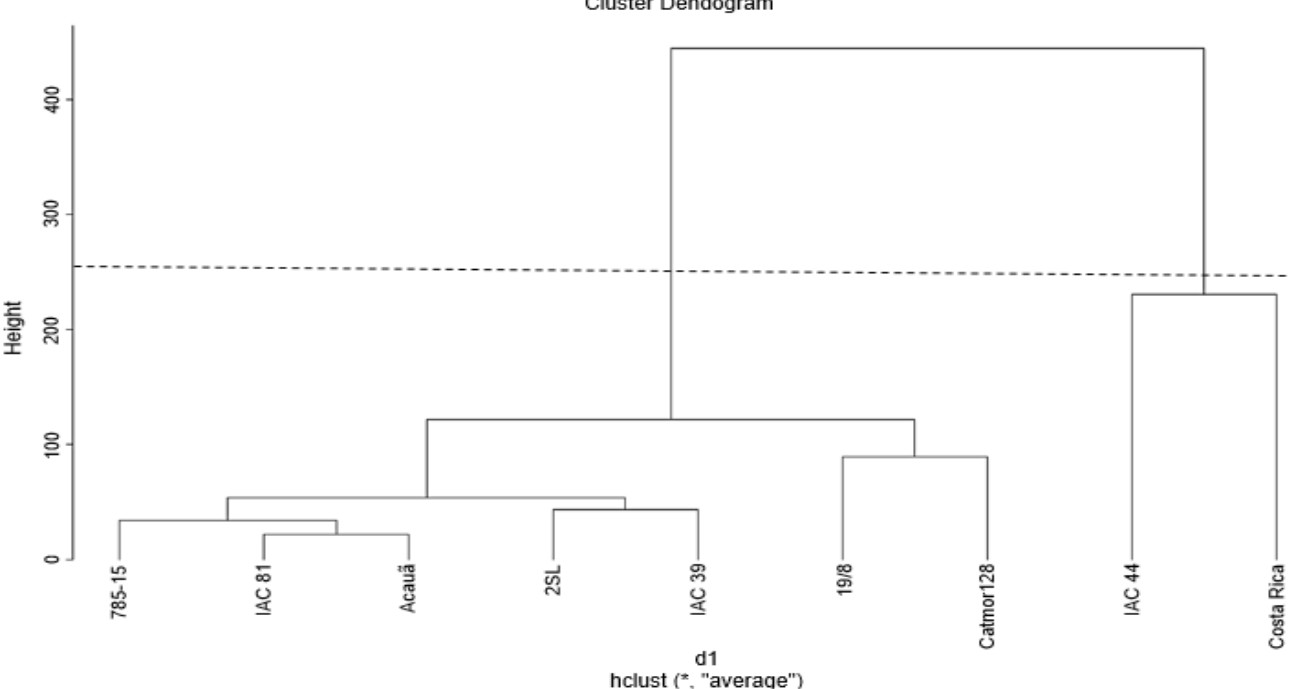

**Figure 1.** Dissimilarity dendrogram for nine *Coffea arabica* cultivars, grouped by UPGMA, using Mahalanobis' generalized distance. Co-phenetic correlation: 0.8332.

Slight differences were observed between the clustering methods, as the cultivars were also divided into two groups using the Tocher method. One group contained only one cultivar (Catuaí Vermelho IAC 44), while the other seven genotypes formed a second group (Table 4). For both UPGMA and Tocher methods, the cultivar Catuaí Vermelho IAC 44 was considered divergent because of its high dissimilarity degree.

**Table 4.** Clusters of the variables of coffee bean, husk, fruits and leaves, and nutrient concentration and accumulation in nine *Coffea arabica* L. cultivars, grown in the Gorongosa Mountain, Gorongosa National Park, Sofala, Mozambique, grouped by the Tocher method and Mahalanobis generalized distance (D2), with an average distance between the clusters of 362.9995.

| Groups | Cultivars |
| --- | --- |
| I | Catuaí Vermelho IAC 81, Acauã, Catuaí Amarelo IAC 39, Catucaí 785-15, Catucaí Amarelo 2SL, Catucaí Vermelho 19/8, Catimor 128, Costa Rica |
| II | Catuaí Vermelho IAC 44 |

The variables analyzed indicated different performances of the cultivars by Tukey's test (Table 5). The coefficient of experimental variation ranged from 2.83% to 6.21%, which indicated good precision in the evaluation of the characteristics. The percentage of bean in the fruit, husk percentage in the fruit, ratio of ripe fruit weight/dry bean weight, ripe fruit weight/ton of bean 11%MC, ripe fruit weight/60 kg green bean 11%MC, and ripe fruit/ton of bean 11%MC did not differ significantly among the cultivars.

**Table 5.** Characteristics of ripe and dried fruit and coffee bean in nine *Coffea arabica* cultivars, grown on the Gorongosa Mountain, Gorongosa National Park, Sofala, Mozambique.

| Cultivars | B | H | DBW | RFW | DFW | RFW/ DFW | RFW/ DBW | Ripe Fruit Weight/ton of Bean | Ripe Fruit Weight/60 kg of Bean | Dry Fruit Weight/ton of Bean |
|---|---|---|---|---|---|---|---|---|---|---|
| | % | % | g | g | G | | | kg$^{-1}$ | kg$^{-1}$ | ton$^{-1}$ |
| Catuaí Vermelho IAC 44 | 62.2 a | 37.8 a | 0.17 ab | 1.60 ab | 0.50 ab | 3.21 b | 5.16 a | 5166 a | 310 a | 1608 a |
| Catucaí Amarelo 2SL | 59.9 a | 40.1 a | 0.13 c | 1.34 c | 0.41 c | 3.24 ab | 5.41 a | 5415 a | 325 a | 1671 a |
| Catuaí Amarelo IAC 39 | 61.5 a | 38.5 a | 0.15 abc | 1.50 bc | 0.47 ab | 3.13 b | 5.09 a | 5096 a | 306 a | 1628 a |
| Catucaí 785-15 | 61.5 a | 38.5 a | 0.15 bc | 1.46 bc | 0.44 bc | 3.24 ab | 5.28 a | 5282 a | 317 a | 1629 a |
| Catucaí Vermelho 19/8 | 60.9 a | 39.1 a | 0.15 bc | 1.54 abc | 0.47 abc | 3.27 ab | 5.39 a | 5398 a | 324 a | 1648 a |
| Catuaí Vermelho IAC 81 | 61.1 a | 38.9 a | 0.16 ab | 1.57 abc | 0.49 ab | 3.15 b | 5.18 a | 5182 a | 311 a | 1640 a |
| Acauã | 62.9 a | 37.1 a | 0.17 ab | 1.64 ab | 0.51 ab | 3.21 b | 5.11 a | 5111 a | 307 a | 1589 a |
| Catimor 128 | 62.5 a | 37.5 a | 0.18 a | 1.77 a | 0.52 a | 3.35 ab | 5.36 a | 5360 a | 322 a | 1601 a |
| Costa Rica | 61.7 a | 38.3 a | 0.15 bc | 1.61 ab | 0.46 abc | 3.44 a | 5.58 a | 5586 a | 335 a | 1623 a |
| CV (%) | 3.74 | 5.99 | 6.21 | 6.17 | 5.41 | 2.83 | 4.73 | 4.73 | 4.73 | 3.76 |

For each parameter, the means followed by different letters indicate significant differences between cultivars (a, b, c). CV: coefficient of variation; B: % bean in the fruit; H: % husk in the fruit; DBW: weight of one bean 11%MC; RFW: ripe fruit weight; DFW: dry fruit weight 11%MC; RFW/DFW: ratio ripe fruit weight/dry fruit weight; RFW/DBW: ratio ripe fruit weight/dry bean weight; Ripe fruit weight per ton of bean: ripe fruit weight/ton of bean 11%MC; Ripe fruit weight per 60 kg bag: weight of ripe fruit per 60 kg of bean 11%MC; Ripe coffee per ton of processed bean: dry coffee per ton of bean 11%MC.

For the weight of one bean 11%MC, ripe fruit weight, and dry fruit weight 11%MC, the cultivars differed significantly. The highest mean proportions (0.18, 1.77, and 0.52, respectively) were always observed for cv. Catimor 128, and the lowest means (0.13, 1.34, and 0.41, respectively) for cv. Catucaí Amarelo 2SL.

A slight variation was observed in the ratio of fruit ripe weight/fruit dry weight. The Costa Rica cultivar obtained the highest mean ratio (3.44), and cv. Catuaí Amarelo IAC 39 the lowest (3.13).

### 3.2. Nutrient Concentration in Bean, Husk, and Leaf

With regard to the bean nutrient concentrations, the macronutrients N, P, and Ca did not differ significantly among the cultivars (Figure 2A). For K and Mg, cv. Costa Rica had the highest concentrations (18 g kg$^{-1}$ and 2.42 g kg$^{-1}$), whereas the other cultivars did not differ from each other (except for K). The mean S concentration was highest (3.25 g kg$^{-1}$) for cv. Costa Rica and lowest (1.72 g kg$^{-1}$) for cv. Catuaí Vermelho IAC 44. In general, the mean concentrations of all macronutrients were the highest (24 g kg$^{-1}$) for cv. Catucaí Vermelho 19/8.

The micronutrient concentrations Cu and Zn in the bean were similar in all cultivars (Figure 2B). For B, cv. Costa Rica contained the highest concentrations (52.1 mg kg$^{-1}$), different from all other cultivars (which did not differ from each other). For Mn, cv. Catimor 128 contained the highest concentrations among all cultivars (92.4 mg kg$^{-1}$), and cv. Catucaí 785-15 the lowest (59.4 mg kg$^{-1}$). For Fe, cv. Catuaí Vermelho IAC 44 stood out with the highest concentration (147.3 mg kg$^{-1}$), cv. Catucaí Vermelho 19/8 had a low mean (66.6 mg kg$^{-1}$), and cv. Catucaí Amarelo 2SL did not differ from the others in terms of B.

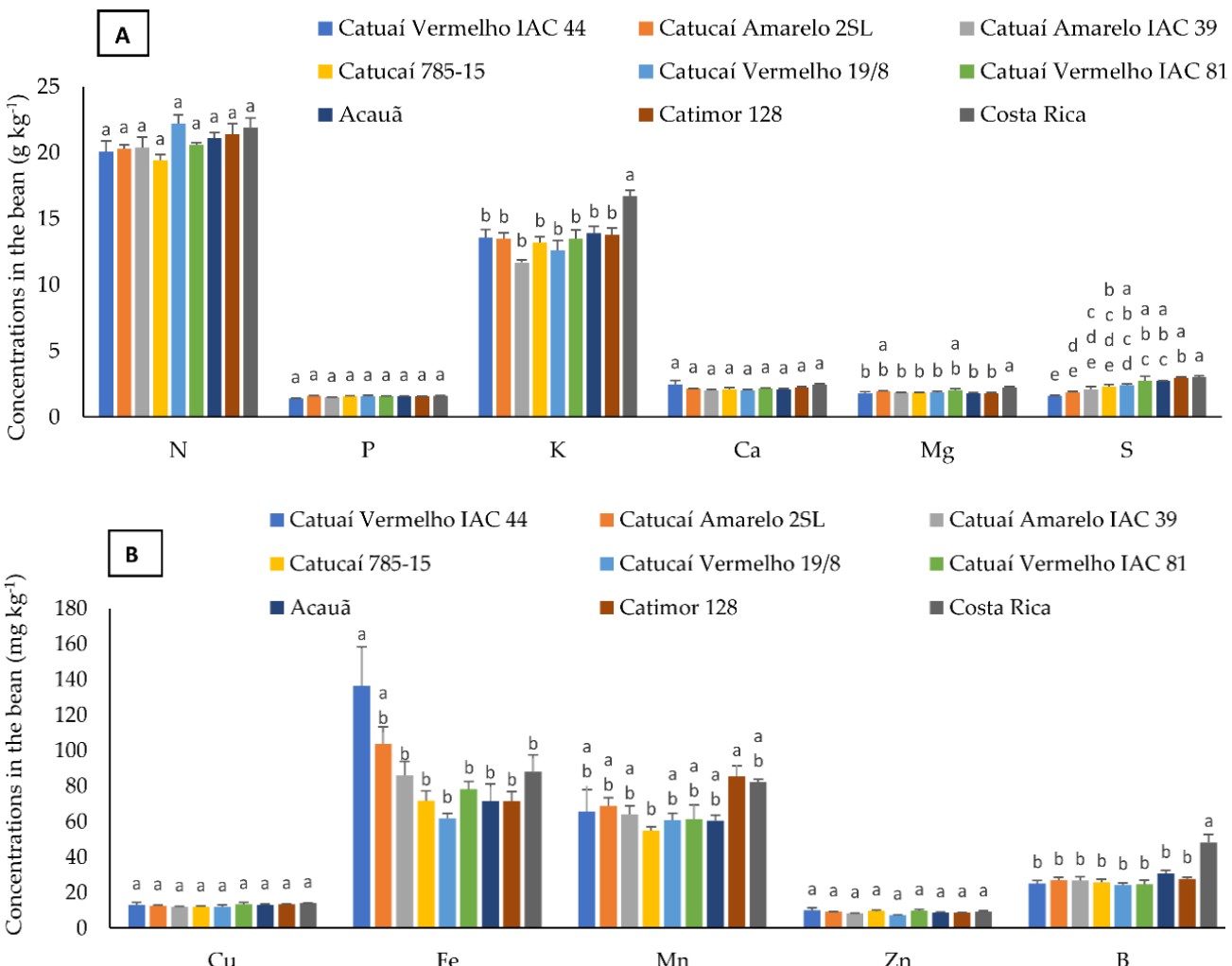

**Figure 2.** Concentration of macronutrients (**A**) and micronutrients (**B**) in the bean of nine *Coffea arabica* cultivars grown in Gorongosa National Park, Mozambique. For each parameter, the mean values $\pm$ SE ($n$ = 4) followed by different letters indicate significant differences between cultivars (a, b, c, d, e).

For the macronutrients K and Mg in the husk, the cultivars did not influence the nutrient concentrations according to the Tukey test (Figure 3A). For N, cv. Catucaí Vermelho 19/8 had the highest mean concentration (16.5 g kg$^{-1}$). The highest mean P concentration was found in cv. Catucaí Amarelo 2SL (1.4 g kg$^{-1}$), and the lowest in cvs. Acauã and Catimor 128 (1.05 and 1.02 g kg$^{-1}$). For Ca, two groups were formed, although some cultivars did not differ significantly from the others. The highest means were found for cvs. Catimor 128 and Costa Rica (6.42 and 6.4 g kg$^{-1}$) and the lowest values for cvs. Catuaí Amarelo IAC 39 and Catucaí 785-15 (5.25 and 5.3 g kg$^{-1}$). For S, two groups were formed as well, and some cultivars did not differ from the others. The cultivar Acauã contained the highest mean concentration (4.35 g kg$^{-1}$), and cv. Catuaí Vermelho IAC 44 the lowest (3.6 g kg$^{-1}$).

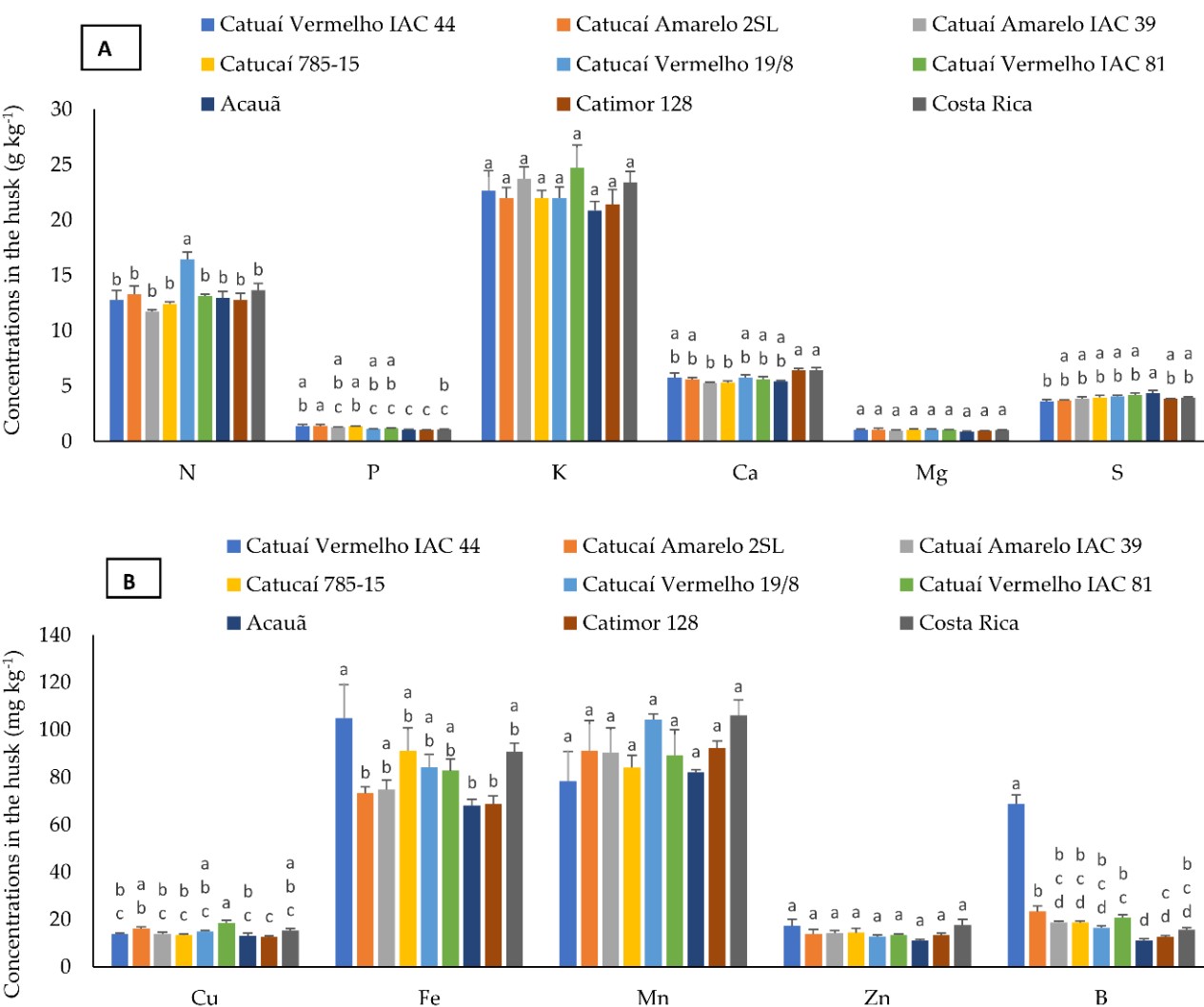

**Figure 3.** Concentration of macronutrients (**A**) and micronutrients (**B**) in the husk of nine *Coffea arabica* cultivars grown in Gorongosa National Park, Mozambique. For each parameter, the mean values $\pm$ SE ($n$ = 4) followed by different letters indicate significant differences between cultivars (a, b, c, d).

The concentrations of the micronutrients Mn and Zn in the husk did not differ among the cultivars, all of which were grouped using Tukey's test (Figure 3B). Iron was clustered in two groups, although some cultivars did not differ significantly from the others. Cultivar Catuaí Vermelho IAC 44 had the highest (104.9 mg kg$^{-1}$), and cvs. Catucaí Amarelo 2SL, Catimor 128, and Acauã exhibited the lowest mean concentrations (73.2, 68.8, and 68.1 mg kg$^{-1}$). The mean Cu concentrations were the highest for cv. Catuaí Vermelho IAC 44 (13.9 mg kg$^{-1}$) and the lowest for cv. Catimor 128 (12.6 mg kg$^{-1}$). For B, cv. Catuaí Vermelho IAC 44 had, by far, the highest mean concentration (68.7 mg kg$^{-1}$), and cv. Acauã the lowest (11.1 mg kg$^{-1}$).

The leaf macronutrient concentrations did not differ among the cultivars in N, Ca, and Mg (Figure 4A). Phosphorus differed only slightly, although some cultivars did not differ significantly from the others. The means were the highest for cvs. Acauã, Catucaí Amarelo 2SL, and Catuaí Vermelho IAC 44 (1.37, 1.37, and 1.35 g kg$^{-1}$, respectively), and lowest for cv. Catimor 128 (1.05 g kg$^{-1}$). For K, the highest means were recorded for cv. Catuaí Vermelho IAC 44 (19.4 g kg$^{-1}$), and the lowest for cv. Catimor 128 (12.4 g kg$^{-1}$). The mean S concentrations in cv. Catimor 128 and Costa Rica were the lowest (1.72 and 1.35 g kg$^{-1}$, respectively), significantly different from all other cultivars.

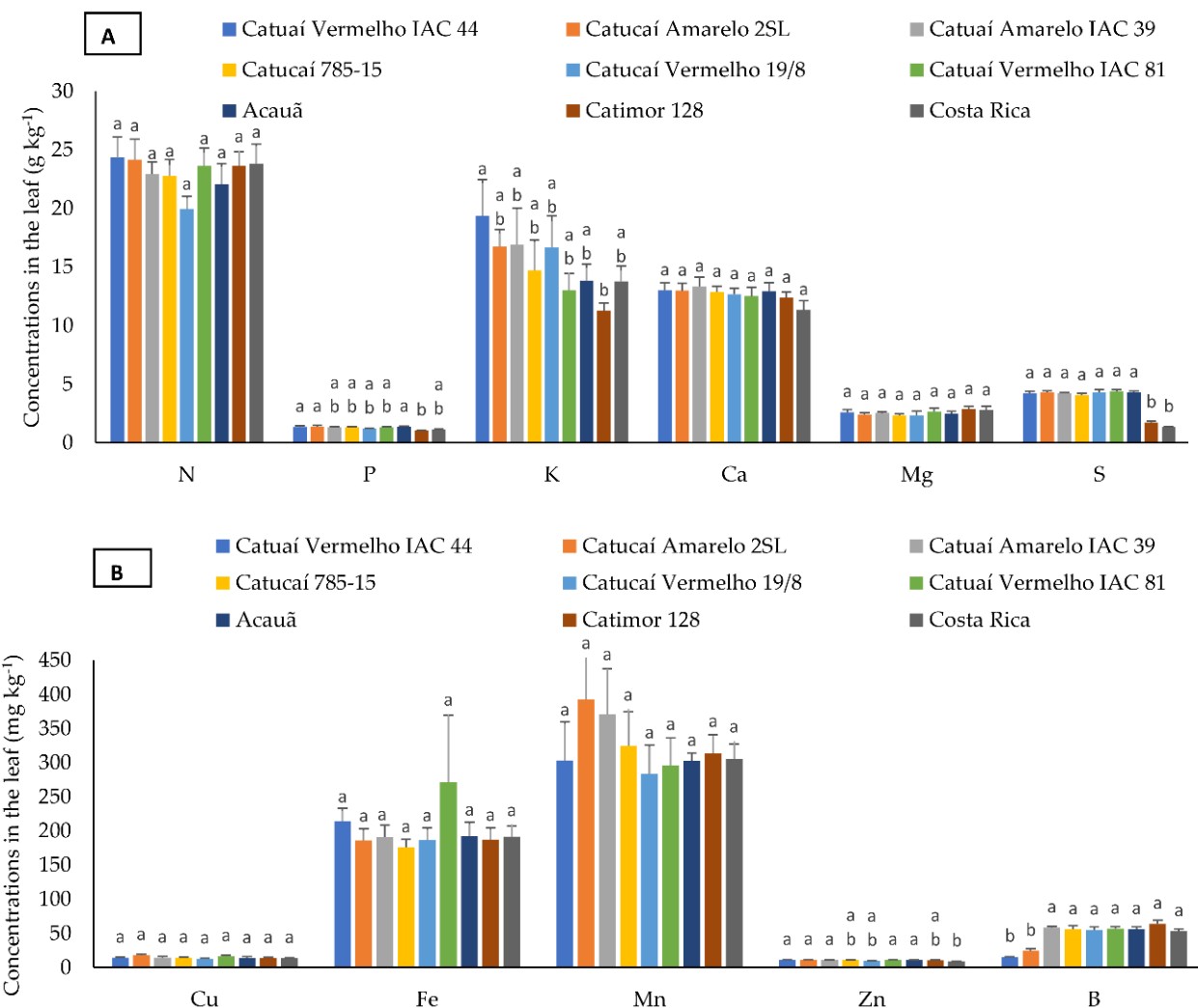

**Figure 4.** Concentration of macronutrients (**A**) and micronutrients (**B**) in the leaves of nine *Coffea arabica* cultivars grown in Gorongosa National Park, Mozambique. For each parameter, the mean values $\pm$ SE (*n* = 4) followed by different letters indicate significant differences between cultivars (a, b).

Regarding leaf micronutrients, no differences were found for Cu, Fe, and Mn among the cultivars (Figure 4B). For Zn, the cultivar means were divided into two groups, although some cultivars did not differ significantly from the others. The highest means were recorded for cv. Catucaí Amarelo 2SL, Catuaí Vermelho IAC 81, Catuaí Vermelho IAC 44, Acauã, and Catuaí Amarelo IAC 39 (11.1, 11, 10.9, 10.9, and 10.7 mg kg$^{-1}$, respectively). The cultivars Catucaí Vermelho 19/8 and Costa Rica were isolated with the lowest means (9.62 mg kg$^{-1}$ and 8.67 mg kg$^{-1}$). For B, the cvs. Catucaí Amarelo 2SL and Catuaí Vermelho IAC 44 presented the lowest concentrations (24.1 and 15.4 mg kg$^{-1}$) in comparison with the other cultivars. The cultivars Catuaí Amarelo IAC 39, Catuaí Vermelho IAC 81, and Acauã were always in the group with the highest micronutrient means.

### 3.3. Nutrient Accumulation in the Fruit

The accumulation of the macronutrients N, P, K, Mg, and S in the fruit differed significantly among the cultivars, according to the Tukey test (Table 6). The highest values were found for cvs. Catucaí Vermelho 19/8 (N), Catucaí Amarelo 2SL (P), and Costa Rica (K, Mg, and S) and the lowest values for cvs. Catucaí 785-15 (N), Acauã (P, K, Mg), and Catuaí Vermelho IAC 44 (S). Calcium was the only macronutrient with no significant differences among the cultivars.

**Table 6.** Nutrients accumulated in the fruit of nine *Coffea arabica* cultivars, grown in the Gorongosa Mountain region, Gorongosa National Park, Sofala, Mozambique.

| Cultivars | Macronutrients | | | | | | Micronutrients | | | | |
|---|---|---|---|---|---|---|---|---|---|---|---|
| | N | P | K | Ca | Mg | S | Cu | Fe | Mn | Zn | B |
| | (g kg$^{-1}$) | | | | | | (mg kg$^{-1}$) | | | | |
| | Fruit | | | | | | | | | | |
| Catuaí Vermelho IAC 44 | 27.3 b | 2.15 ab | 26.3 ab | 5.74 a | 2.40 ab | 3.62 b | 20.8 a | 195.5 a | 110.3 a | 19.9 a | 63.4 a |
| Catucaí Amarelo 2SL | 28.5 ab | 2.46 a | 27.1 ab | 5.63 a | 2.59 ab | 4.17 ab | 22.7 a | 149.2 ab | 125.1 a | 17.7 a | 41.3 b |
| Catuaí Amarelo IAC 39 | 27.2 b | 2.22 ab | 25.5 b | 5.06 a | 2.38 ab | 4.32 ab | 20.1 a | 129.2 b | 116.3 a | 16.7 a | 37.6 b |
| Catucaí 785-15 | 26.7 b | 2.35 ab | 26.0 ab | 5.19 a | 2.43 ab | 4.60 ab | 19.8 a | 124.8 b | 103.9 a | 17.8 a | 36.5 b |
| Catucaí Vermelho 19/8 | 32.1 a | 2.22 ab | 25.8 b | 5.48 a | 2.47 ab | 4.84 ab | 20.9 a | 112.8 b | 123.0 a | 15.0 a | 34.0 b |
| Catuaí Vermelho IAC 81 | 28.6 ab | 2.24 ab | 27.9 ab | 5.41 a | 2.63 ab | 5.24 a | 24.1 a | 127.8 b | 115.3 a | 17.7 a | 36.7 b |
| Acauã | 28.1 ab | 2.12 b | 25.3 b | 5.03 a | 2.26 b | 5.11 a | 20.4 a | 108.8 b | 105.3 a | 14.6 a | 36.8 b |
| Catmor 128 | 28.5 ab | 2.14 ab | 25.6 b | 5.81 a | 2.31 ab | 5.08 a | 20.2 a | 109.8 b | 137.1 a | 16.2 a | 34.7 b |
| Costa Rica | 29.7 ab | 2.21 ab | 30.1 a | 6.10 a | 2.82 a | 5.28 a | 22.8 a | 140.2 ab | 143.1 a | 19.2 a | 57.3 a |
| CV (%) | 6.41 | 6.09 | 6.52 | 9.4 | 8.65 | 10.98 | 8.96 | 18.4 | 18.0 | 14.2 | 10.5 |

For each parameter, the means followed by different letters indicate significant differences between cultivars (a, b).

Major differences in the micronutrient accumulation in the fruits were observed for Fe and B, with maximum values of both minerals in cv. Catuaí Vermelho IAC 44. Regarding Cu, Mn, and Zn, these minerals did not differ significantly among the cultivars. The highest mean for the micronutrients was observed for cv. Catuaí Vermelho IAC 44 (195.5 mg kg$^{-1}$), and the lowest values were observed for cv. Acauã (14.6 mg kg$^{-1}$).

*3.4. Correlations between Cumulative Nutrient Concentrations in Bean, Husk, Fruit, and Husk Percentage, Bean Percentage, Bean Weight, and Fruit Weight*

Spearman's correlations for macro- and micronutrient concentrations of 30% were considered significant, with binary relationships (positive and negative) between the evaluated variables (Table 7). However, only significantly positive correlations were observed.

Among the macronutrients, N was very strongly correlated with Bean × NAB, Husk × NAF, and Fruit weight × Bean weight. Strong correlations were observed for Bean × NAF, Husk × NAH, NAF × NAH, %Bean× Fruit weight, %Bean × Bean weight, and Fruit weight × Bean weight, whereas moderate correlations were identified for Bean × Husk and NAH × %Husk.

For P, very strong correlations were observed for Bean × NAB, and Husk × NAH, and strong correlations for Husk × NAF, NAF × NAH, and NAF × %Husk. Moderate correlations were also found for Husk × %Husk and NAH × %Husk.

Potassium had very strong correlations with Bean × NAB, strong correlations with Husk × NAH and NAH × %Husk, and moderate correlations with Bean × %Husk and NAH × Bean weight.

Calcium was very strongly correlated with Bean × NAB, Husk × NAF, and NAF × NAH. Strong correlations were observed for Bean × Husk, Bean × NAF, and Husk × NAH.

Magnesium presented very strong correlations with Bean × NAB, Bean × NAF, and NAH × %Husk, as well as strong correlations with Husk × NAH, and moderate correlations for Bean × NAH, Bean × %Husk, Husk × %Husk, NAF × NAH, and NAF × %Husk.

Sulphur was shown to have very strong correlations with Bean × NAB and Bean × NAF, as well as strong positive correlations with Husk × NAH and Husk × NAF.

With regard to the micronutrients, Cu exhibited very strong correlations with the variables Bean × NAB and Husk × NAH, but also strong correlations with Husk × %Husk, Husk × NAF, NAF × NAH, and NAH × %Husk.

For Fe, only very strong correlations were recorded for Bean × NAB, Bean × NAF, and Husk × NAH.

Manganese was shown to have very strong correlations with Bean × NAB, strong correlations with Bean × NAF, Husk × NAH, and Husk × NAF, and moderate correlations with NAF × NAH and NAH × %Husk.

Very strong correlations were observed for Zn with Bean × NAB, Husk × NAH, and Husk × NAF, whereas strong correlations were depicted for Bean × NAF and NAF × NAH. A moderate correlation was also found with Bean × Husk.

The micronutrient B was very strongly correlated with Bean × NAB and Husk × NAH.

Finally, it is noteworthy that the variables Bean × NAB and Husk × NAH commonly presented very strong or strong correlations with all macro- and micronutrients.

**Table 7.** Spearman correlation matrix between nutrient accumulation in bean, husk, fruit, husk percentage, bean percentage, fruit weight, and bean weight in nine *Coffea arabica* cultivars, grown in the Gorongosa Mountain region, Gorongosa National Park, Sofala, Mozambique.

| Variables | Nutrients | | | | | | | | | | |
|---|---|---|---|---|---|---|---|---|---|---|---|
| | N | P | K | Ca | Mg | S | Cu | Fe | Mn | Zn | B |
| Bean × Husk | 0.64 * | −0.23 | −0.30 | 0.71 * | 0.38 | 0.54 | 0.15 | 0.38 | 0.50 | 0.6 * | −0.55 |
| Bean × NAB | 1 *** | 1 *** | 1 *** | 1 *** | 1 *** | 1 *** | 0.99 *** | 1 *** | 1 *** | 1 *** | 1 *** |
| Bean × NAH | 0.28 | −0.18 | −0.49 | 0.55 | 0.60 * | 0.27 | 0.00 | 0.42 | 0.22 | 0.48 | −0.56 |
| Bean × NAF | 0.82 ** | 0.18 | 0.23 | 0.80 ** | 0.96 *** | 0.95 *** | 0.57 | 0.93 *** | 0.83 ** | 0.83 ** | 0.35 |
| Bean × %Bean | 0.09 | −0.18 | 0.66 * | 0.30 | −0.69 * | 0.31 | 0.40 | −0.12 | 0.12 | 0.08 | 0.55 |
| Bean × %Husk | −0.09 | 0.18 | −0.66 * | −0.30 | 0.69 * | −0.31 | −0.40 | 0.12 | −0.12 | −0.08 | −0.54 |
| Bean × Fruit weight | 0.29 | −0.35 | 0.37 | 0.11 | −0.65 * | 0.29 | 0.33 | −0.36 | 0.12 | −0.06 | 0.05 |
| Bean × Bean weight | 0.14 | −0.31 | 0.47 | 0.29 | −0.66 * | 0.30 | 0.44 | −0.28 | 0.11 | 0.18 | 0.09 |
| Husk × %Husk | 0.39 | 0.60 * | 0.39 | −0.24 | 0.64 * | −0.06 | 0.75 ** | 0.18 | 0.34 | 0.04 | 0.54 |
| Husk × NAH | 0.85 ** | 0.99 *** | 0.83 ** | 0.80 ** | 0.86 ** | 0.85 ** | 0.98 *** | 0.98 *** | 0.87 ** | 0.92 *** | 1 *** |
| Husk × NAF | 0.91 *** | 0.75 ** | 0.56 | 0.90 *** | 0.55 | 0.71 * | 0.83 ** | 0.55 | 0.88 ** | 0.93 *** | 0.47 |
| Husk × %Bean | −0.39 | −0.6 * | −0.39 | 0.24 | −0.64 * | 0.06 | −0.75 ** | −0.18 | −0.34 | −0.04 | −0.54 |
| Husk × Bean weight | −0.29 | −0.60 * | −0.29 | 0.39 | −0.44 | 0.12 | −0.59 * | −0.20 | −0.39 | −0.31 | −0.33 |
| Husk × Fruit weight | −0.22 | −0.66 * | −0.27 | 0.34 | −0.52 | 0.16 | −0.54 | −0.35 | −0.29 | −0.47 | −0.37 |
| NAF × NAH | 0.73 * | 0.80 ** | 0.52 | 0.9 ** | 0.68 * | 0.47 | 0.77 ** | 0.58 | 0.68 * | 0.86 ** | 0.44 |
| NAF × %Bean | −0.3 | −0.90 *** | −0.43 | 0.03 | −0.65 * | 0.24 | −0.38 | −0.35 | −0.17 | −0.09 | 0.14 |
| NAF × %Husk | 0.3 | 0.80 ** | 0.43 | −0.03 | 0.65 * | −0.24 | 0.38 | 0.35 | 0.17 | 0.09 | −0.14 |
| NAF × Fruit weight | 0.01 | −0.83 ** | −0.49 | −0.03 | −0.64 * | 0.22 | −0.13 | −0.48 | −0.11 | −0.40 | −0.19 |
| NAF × Bean weight | −0.097 | −0.80 ** | −0.36 | 0.06 | −0.60 * | 0.22 | −0.18 | −0.42 | −0.19 | −0.19 | −0.14 |
| NAH × %Bean | −0.66 * | −0.63 * | −0.75 ** | −0.23 | −0.90 *** | −0.44 | −0.84 ** | −0.24 | −0.69 * | −0.26 | −0.55 |
| NAH × %Husk | 0.66 * | 0.63 * | 0.75 ** | 0.23 | 0.90 *** | 0.44 | 0.84 ** | 0.24 | 0.69 * | 0.26 | 0.55 |
| NAH × Fruit weight | −0.49 | −0.70 * | −0.54 | −0.18 | −0.70 * | −0.19 | −0.54 | −0.41 | −0.50 | −0.58 | −0.39 |
| NAH × Bean weight | −0.49 | −0.62 * | −0.62 * | −0.15 | −0.68 * | −0.25 | −0.62 * | −0.25 | −0.63 * | −0.49 | −0.33 |
| %Bean × %Husk | −1 * | | | | | | | | | | |
| %Bean × Fruit weight | 0.73 * | | | | | | | | | | |
| %Bean × Bean weight | 0.80 ** | | | | | | | | | | |
| %Husk × Fruit weight | −0.73 * | | | | | | | | | | |
| %Husk × Bean weight | −0.80 ** | | | | | | | | | | |
| Fruit weight × Bean weight | 0.96 *** | | | | | | | | | | |

Values in blue and red mean significant positive and negative correlations, respectively * $p$ <0.05, ** $p$ <0.01, *** $p$ <0.001. %Bean: bean percentage; %Husk: husk percentage; NAB: total nutrient accumulation in beans; NAH: total nutrient accumulation in the husk; NAF: total nutrient accumulation in the fruits.

## 4. Discussion

Two groups of cultivars were formed by the UPGMA (Figure 1) and two by the Tocher (Table 4) method, using the dissimilarity between the fruit traits and the macro- and micronutrient concentrations and accumulation in the leaf and fruit of the nine cultivars (Figure 1). The second method indicated that one of groups integrated only one cultivar, suggesting that this genotype differs significantly from all of the others with regard to the studied traits [36,42,43]. This hierarchical method was previously used [17,25] to assess genetic diversity, showing the formation of divergent groups of *C. canephora* and *C. arabica* based on nutritional characteristics; that is, the traits are expressed (and result) from the existing genetic diversity, which is extremely important for breeding.

Coffee fruits are morphologically characterized by the following parts: a pericarp (exocarp, mesocarp, and endocarp), perisperm, and endosperm, of which the latter constitutes storage tissue where the embryo is located [44,45]. Post-harvest processing eliminates part of these fruit components, resulting in a mixture of residues that can be re-applied as husk mulch to the crop in order to return nutrients to the soil, thus decreasing the use of chemical fertilizers. The higher or lower percentage of bean in the fruit indicated genotypic

variation in the biomass allocation to the different fruit constituents. Unlike the others, cvs. Catimor 128, Catuaí Vermelho IAC 44, Acauã, and Catuaí Vermelho IAC 81 accumulate a higher percentage of the bean in each fruit, which is the commercially exploited part, consequently reducing the energy in the form of biomass in the other fruit parts. Dry matter accumulation in the fruit is influenced by the productivity of the ripe fruit; in other words, a higher ripe fruit yield results in more biomass accumulated in the fruit. The yield of green coffee is measured after the drying and depulping of the coffee fruit, and the greater or lesser weight of green coffee bean fruit is directly related to the biomass accumulation in the fruit [45,46].

Low variability was observed among cultivars regarding the weight of ripe fruit per 60 kg of bean 11%MC and dry coffee per ton of bean 11%MC. The productivity ranged from 306 to 335 kg$^{-1}$ of ripe fruit to obtain a 60 kg bag, representing a coefficient of variation of 4.73% between the evaluated cultivars, where CVs below 7% are generally considered low [46].

From the studied macro- and micronutrients, N, K, and Ca, and Fe, Mn, and B stood out with highest accumulation in the bean, husk, fruit, and leaves, confirming previous findings for *C. canephora* cv. Conilon [20]. Among the micronutrients, Mn was the second most accumulated. Approximately 80% of Fe is accumulated in chloroplasts and plays a vital role in photosynthesis and chlorophyll production [19,37]. Nutrient concentration varies according to the soil conditions, season, plant age, and genotype maturation cycle [23,26]. In addition, some genotypes take up and translocate more nutrients than others [47]. Moreover, the efficiency and priority of uptake of some nutrients are genetically defined characteristics [29]. Several studies describe differences in the nutritional concentrations between *C. arabica* genotypes under the same management conditions [30,31]. This is related to a wide interspecific variability between *C. arabica* genotypes, especially in relation to characteristics such as growth, maturation cycle, nutrient accumulation, and stress tolerance [32]. This makes it possible to explore diversity for the identification and breeding of genetic material better adapted to the range of soil and climatic conditions of the producing regions [33]. Additionally, all studied minerals (both macro- and micro-nutrients) significantly correlated with the Bean × NAB and Husk x NAH variables, suggesting that these can be used to follow the mineral balance and help crop fertilization management. Nevertheless, for bean quality purposes, it should be pointed out that physical and chemical traits (including bean size and color, size, flavor, phenolic content, soluble solids, chlorogenic, caffeic and *p*-coumaric acids, caffeine, trigonelline, and lipids), but also mineral content, can be considerably impacted by higher temperatures (although less under elevated [$CO_2$] [7,11–13]), and must be additionally considered when breeding new cultivars to cope with future expected climate conditions.

## 5. Conclusions

Genetic divergence in *C. arabica* cultivars was detected in the ripe and dried fruit and in the coffee bean with regard to the assessed variables.

The UPGMA detected the dissimilarity of cvs. Costa Rica and Catuaí Vermelho IAC 44 in relation to the others, whereas the Tocher method pointed to the dissimilarity of cv. Catuaí Vermelho IAC 44 in relation to the others.

The evaluated cultivars showed higher accumulation in most macronutrients and some micronutrients in the coffee bean, fruit, and husk. The macronutrients N, K, and Ca were the most accumulated in the bean, husk, fruit, and leaves, and the micronutrients Fe, Mn, and B were the most accumulated in the bean, husk, fruit, and leaves of *C. arabica*.

The results of this study may underlie the development of future coffee cultivars for recommendation in Mozambique as a new frontier of coffee cultivation, and future research is suggested to find develop region-specific lines for different environments.

**Author Contributions:** Conceptualization, F.L.P., A.I.R.-B., W.P.R. and J.C.R.; formal analysis, N.J.A. and F.L.P.; funding acquisition, J.C.R., A.I.R.-B. and F.L.P.; investigation, N.J.A., F.L.P., Z.A. and S.S.M.; methodology, C.A.K. and F.L.P.; project administration, F.L.P., A.I.R.-B. and J.C.R.; supervision, F.L.P.; visualization, N.J.A., F.L.P., J.C.R., C.A.K. and A.P.V.; writing—original draft, N.J.A., F.L.P., W.P.R. and A.P.V.; writing—review and editing, all authors. All authors have read and agreed to the published version of the manuscript.

**Funding:** This study was supported by the national funds of Camões—Institute for Cooperation and Language (Portugal), the Brazilian Cooperation Agency (Brazil), and the National Park of Gorongosa (Mozambique), through the Project Gorongosa Café (TriCafé). the Foundation for Support of Research and Innovation of Espírito Santo (FAPES). Portuguese national funding support was also provided by the Foundation for Science and Technology, I.P. (FCT), through the research units UIDB/00239/2020 (CEF) and UIDP/04035/2020 (GeoBioTec), and Associate Laboratory TERRA (LA/P/0092/2020). A grant from CNPq, Brazil, to F. Partelli is also greatly acknowledged.

**Data Availability Statement:** Data are contained within the article.

**Acknowledgments:** The Federal University of Espírito Santo—UFES, the Graduate Program in Genetics and Breeding—PPGGM, University of Lisbon, Gorongosa National Park, Conilon Coffee Research Excellence Core, the National Council for Scientific and Technological Development—CNPq, the Brazilian Federal Agency for Support and Evaluation of Graduate Education—CAPES, and the Support Foundation for Research and Innovation of Espírito Santo—FAPES for funding.

**Conflicts of Interest:** The authors declare no conflict of interest.

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
