# Peer review of "Diversity in Coffea arabica Cultivars in the Mountains of Gorongosa National Park, Mozambique, Regarding Bean and Leaf Nutrient Accumulation and Physical Fruit Traits"

_agronomy, doi:10.3390/agronomy13041162_

Round 1
Reviewer 1 Report
The authors evaluated the diversity in fruit traits and nutrient accumulation in different Coffea arabica cultivars cropped in the Gorongosa National Park - Mozambique. They used a randomized block design to compare nine coffee cultivars and collected fruit and leaf samples over two months. They analyzed the physical and chemical characteristics of the fruit, husk, bean and leaves, and clustered the cultivars by their similarity. They found significant differences in the performance of the cultivars, as well as in the nutrient concentrations and accumulation in the different organs. They concluded that the genetic diversity and evaluation period must be taken into account for a nutritional crop management and diagnosis that ensures high yields and quality. They also highlighted the potential of the region for coffee production in agroforestry systems.
1. Catucai Vermelho IAC81 has a greater Fe concentration than other cultivars, as shown in Figure 4B. However, no significant difference was revealed by the statistical analysis. With a comparison of roughly 250 to 150 mg/kg, this sort of outcome is unusual. The error bars seemed the same in several figures as well. Please double-check the data and provide the raw data for each figure in the manuscript, together with the mean and standard deviation.
2. Line 47 “10,2 million tons” should be “10.2 million tons”
3. Line 77 and Line 458 “CO2” should be “CO2”
4. The quality of Figure 3 and Figure 4 is low.
Author Response
Response to Reviewer 1 Comments
We are thanks for your valuable revision in our manuscript. Considering the time of 7 days proposed to minor revision indicated by the journal, we worked to make the maximum of changes meeting your suggestions. We appreciate all suggestions and are open to new contributions.
About the suggestions and corrections, we numbering to easy our responses:
Point 1: Catucai Vermelho IAC81 has a greater Fe concentration than other cultivars, as shown in Figure 4B. However, no significant difference was revealed by the statistical analysis. With a comparison of roughly 250 to 150 mg/kg, this sort of outcome is unusual. The error bars seemed the same in several figures as well. Please double-check the data and provide the raw data for each figure in the manuscript, together with the mean and standard deviation.
Response 1: We understood the idea about, already checked and made available in annex the tables and respective figures.
Point 2: Line 47 “10,2 million tons” should be “10.2 million tons”.
Response: We understood the idea. We changed to “10.2 million tons”. (lines 47).
Point 3: Line 77 and Line 458 “CO2” should be “CO2”.
Response: We understood the idea, We changed to “CO2”. (lines 77).
Point 4: The quality of Figure 3 and Figure 4 is low.
Response: We understood the idea, We changed to adequate form figure 2, 3 and 4.
Response: We revised the formatting of the manuscript and adjusted references and other aspects of journal requirements. we do not detail the analysis procedures because it is widely known routine procedures and we make the text even longer without need. We do not put the standard error of the mean, because they are in the tables. Even so, at the discretion of the editor we put it.

Reviewer 2 Report
The objective of this study was to evaluate fruit traits and nutrient accumulation in the fruit, husk, and bean, as well as in leave of different coffee cultivars. This is a decent paper but the methodology is completely lacking. The mineral determination is a central part of this work, and yet, the methods are not described at all (a citation is not enough). Not only should the methods be described, the instrumentation should also be described in the experimental section. There are also some inconsistencies in the moisture determination methodology.
Specific comments:
Abstract, lines 31-32: ANOVA and Tukey’s test need not be mentioned in an abstract; these should be discussed in the paper and in the experimental section. Those tests are a means, not an end. Treating them as an end and leaning on these in the abstract makes it seem like you really don’t have anything else to present.
Line 144: Incorrect symbol is used for degrees.
Lines 145-150: Sometimes the authors put a space between a number’s % sign and the units (example: % G, line 146) and sometimes there is no space (example: %MC, line 145). I would suggest coming up with a standard way of denoting %s.
Line 155: Why was the temperature of the convection oven changed? It is 65°C in line 144 but 60°C in line 155.
157-160: “The mineral concentration in the collected materials (fruit, leaf and soil) was quanti-157 fied separately for bean, husk, leaf and soil contents of nitrogen (N), phosphorus (P), po-158 tassium (K), calcium (Ca), magnesium (Mg), sulfur (S), iron (Fe), zinc (Zn), copper (Cu), 159 manganese (Mn) and boron (B), according to the previously described methodology [34].” This methodology should be briefly described, especially since this is not a readily available journal article. Not only should the methodology be outlined, since this is a central analysis in the paper, but the instrumentation should also be presented in the experimental section.
Lines 166-168: It appears at least two different methods are used for moisture analysis, at different temperatures, with different nomenclatures. Why were two different methods used, or two different sets of conditions?
Lines 218-221: I think the authors are confusing correlation and causation. Line 221 states that certain macronutrients contribute to genetic diversity. I think they mean to say that genetic diversity is responsible for high variability in those 5 factors, not that the factors themselves contribute to genetic diversity. It is unlikely that they could tie any of these factors as a cause of genetic diversity. I think I understand what they are trying to say, but genetic diversity contributes to variability in these factors, not the other way around.
Line 529: The number 13 appears to be bolded and in a different font than the rest of the text.
Author Response
Response to Reviewer 2 Comments
We are thanks for your valuable revision in our manuscript. Considering the time of 7 days proposed to minor revision indicated by the journal, we worked to make the maximum of changes meeting your suggestions. We appreciate all suggestions and are open to new contributions.
About the suggestions and corrections, we numbering to easy our responses:
Point 1: Abstract, lines 31-32: ANOVA and Tukey’s test need not be mentioned in an abstract; these should be discussed in the paper and in the experimental section. Those tests are a means, not an end. Treating them as an end and leaning on these in the abstract makes it seem like you really don’t have anything else to present.
Response 1: We understood the idea, We changed to a more adequate form.
Point 2: Line 144: Incorrect symbol is used for degrees.
Response: We understood the idea, We changed to adequate form.
Point 3: Sometimes the authors put a space between a number’s % sign and the units (example: % G, line 146) and sometimes there is no space (example: %MC, line 145). I would suggest coming up with a standard way of denoting %s. (Lines 145-150)
Response 3: We understood the idea, already checked and changed to one form (%s).
Point 4: Why was the temperature of the convection oven changed? It is 65°C in line 144 but 60°C in line 155. (Line 155)
Response 4: We understood the idea, already checked and change to one 65°C.
Point 5: “The mineral concentration in the collected materials (fruit, leaf and soil) was quanti-157 fied separately for bean, husk, leaf and soil contents of nitrogen (N), phosphorus (P), po-158 tassium (K), calcium (Ca), magnesium (Mg), sulfur (S), iron (Fe), zinc (Zn), copper (Cu), 159 manganese (Mn) and boron (B), according to the previously described methodology [34].” This methodology should be briefly described, especially since this is not a readily available journal article. Not only should the methodology be outlined, since this is a central analysis in the paper, but the instrumentation should also be presented in the experimental section. (157-160)
Response 5: We understood the idea, already checked and described methodology used.
Point 6: It appears at least two different methods are used for moisture analysis, at different temperatures, with different nomenclatures. Why were two different methods used, or two different sets of conditions? (Lines 166-168)
Response 6: We understood the idea, We used a method described in only one temperature.
Point 7: I think the authors are confusing correlation and causation. Line 221 states that certain macronutrients contribute to genetic diversity. I think they mean to say that genetic diversity is responsible for high variability in those 5 factors, not that the factors themselves contribute to genetic diversity. It is unlikely that they could tie any of these factors as a cause of genetic diversity. I think I understand what they are trying to say, but genetic diversity contributes to variability in these factors, not the other way around. (Lines 218-221)
Response 7: We understood the idea, already checked and changed like the genetic diversity contributes to variability in these factors listed.
Point 8: The number 13 appears to be bolded and in a different font than the rest of the text. (Line 529)
Response 8: We understood the idea about, already checked and changed to more adequate form.
Response: We revised the formatting of the manuscript and adjusted references and other aspects of journal requirements. we do not detail the analysis procedures because it is widely known routine procedures and we make the text even longer without need. We do not put the standard error of the mean, because they are in the tables. Even so, at the discretion of the editor we put it.

Round 2
Reviewer 1 Report
Please provide the raw data for Figure 2-4 (values of each repeats, not just their means, as in Supplementary Table 1-3). For example, N in Catuaí Vermelho IAC 44 leaves: repeat1: XX.XX g/kg; repeat2: XX.XX g/kg; repeat3: XX.XX g/kg; etc.
These data are not required to be in the paper. As it's only for review, it won't make the manuscript too long.
Author Response
Answers to Reviewer 1 comments and suggestions
The authors evaluated the diversity in fruit traits and nutrient accumulation in different Coffea arabica cultivars cropped in the Gorongosa National Park - Mozambique. They used a randomized block design to compare nine coffee cultivars and collected fruit and leaf samples over two months. They analyzed the physical and chemical characteristics of the fruit, husk, bean and leaves, and clustered the cultivars by their similarity. They found significant differences in the performance of the cultivars, as well as in the nutrient concentrations and accumulation in the different organs. They concluded that the genetic diversity and evaluation period must be taken into account for a nutritional crop management and diagnosis that ensures high yields and quality. They also highlighted the potential of the region for coffee production in agroforestry systems.
Answer: We wish to thank the reviewer for his/her time spent and for the positive comments, suggestions, and corrections. We introduced the suggested modifications.
- Catucai Vermelho IAC81 has a greater Fe concentration than other cultivars, as shown in Figure 4B. However, no significant difference was revealed by the statistical analysis. With a comparison of roughly 250 to 150 mg/kg, this sort of outcome is unusual. The error bars seemed the same in several figures as well. Please double-check the data and provide the raw data for each figure in the manuscript, together with the mean and standard deviation.
Answer:
We have double-checked all statistical analysis. The lack of significance in the mentioned case is related with an (exceptional) large variation between the replicates. For a better perception of such samples variability, we added in all graphs the standard error (as usual in our studies, instead of standard deviation).
Finally, with the clarification of the errors we believe that there is no need to provide the raw data of each figure (since it is not usual). Nevertheless, if the reviewer feels that as mandatory we can provide an excel sheet with the individual replicates and place the data as supplementary data. Please let us know.
- Line 47“10,2 million tons” should be “10.2 million tons”.
Answer: We changed 10,2 to 10.2
- Line 77 and Line 458 “CO2” should be “CO2”.
Answer: We changed CO2 to CO2
- The quality of Figure 3 and Figure 4 is low.
Answer: We provide new figures 2, 3 and 4 with greater quality
Additionally, we revised the formatting of the manuscript and performed minor adjustments some references and other aspects according to the journal requirements.

Reviewer 2 Report
The objective of this study was to evaluate fruit traits and nutrient accumulation in the fruit, husk, and bean, as well as in leave of different coffee cultivars. This is a decent paper but the methodology is completely lacking. The mineral determination is a central part of this work, and yet, the methods are not described at all (a citation is not enough). Not only should the methods be described, the instrumentation should also be described in the experimental section. There are also some inconsistencies in the moisture determination methodology.
Furthermore, it is somewhat alarming that the authors chose not to address the majority of comments raised in the first review of this paper.
Specific comments:
Abstract, lines 33-34: Statistical methods need not be mentioned in an abstract; these should be discussed in the paper and in the experimental section. Those tests are a means, not an end. Treating them as an end and leaning on these in the abstract makes it seem like you really don’t have anything else to present.
Line 164: Why was the temperature of the convection oven changed? It is 65°C in line 146 but 60°C in line 164.
167-177: “The mineral concentration in the collected materials (fruit, leaf and soil) was quantified separately for bean, husk, leaf…. according to the previously described methodology [34].”
This methodology should be briefly described, especially since this is not a readily available journal article. Not only should the methodology be outlined, since this is a central analysis in the paper, but the instrumentation should also be presented in the experimental section.
Lines 187-189: It appears at least two different methods are used for moisture analysis, at different temperatures, with different nomenclatures. Why were two different methods used, or two different sets of conditions?
Line 239: Change “contribution in genetic diversity” to “contribution of genetic diversity.”
Author Response
Answer to Reviewer 2 comments and suggestions
The objective of this study was to evaluate fruit traits and nutrient accumulation in the fruit, husk, and bean, as well as in leave of different coffee cultivars. This is a decent paper but the methodology is completely lacking. The mineral determination is a central part of this work, and yet, the methods are not described at all (a citation is not enough). Not only should the methods be described, the instrumentation should also be described in the experimental section. There are also some inconsistencies in the moisture determination methodology.
Answer: We wish to thank the reviewer for his/her time spent and for the positive comments, suggestions, and corrections, which helped to improve the manuscript. We introduced the suggested modifications as described below.
- Abstract, lines 31-32: ANOVA and Tukey’s test need not be mentioned in an abstract; these should be discussed in the paper and in the experimental section. Those tests are a means, not an end. Treating them as an end and leaning on these in the abstract makes it seem like you really don’t have anything else to present.
Answer: We performed the suggested modification and withdraw the mention to ANOVA and Tukey’s test in the abstract.
- Line 144: Incorrect symbol is used for degrees.
Answer: The change was performed.
- Lines 145-150: Sometimes the authors put a space between a number’s % sign and the units (example: % G, line 146) and sometimes there is no space (example: %MC, line 145). I would suggest coming up with a standard way of denoting %s.
Answer: The change was performed in the entire manuscript, to the form without the space
- Line 155: Why was the temperature of the convection oven changed? It is 65°C in line 144 but 60°C in line 155.
Answer: Thanks for noting this. It was checked and changed to 65 °C.
- 157-160: “The mineral concentration in the collected materials (fruit, leaf and soil) was quantified separately for bean, husk, leaf and soil contents of nitrogen (N), phosphorus (P), potassium (K), calcium (Ca), magnesium (Mg), sulfur (S), iron (Fe), zinc (Zn), copper (Cu), manganese (Mn) and boron (B), according to the previously described methodology [34].” This methodology should be briefly described, especially since this is not a readily available journal article. Not only should the methodology be outlined, since this is a central analysis in the paper, but the instrumentation should also be presented in the experimental section.
Answer: These analytical methods are quite classic, but we recognize the merit of this comment. In this way we provided some additional information (and the relevant references) that we believe will be enough to clarify what is demanded.
- Lines 166-168: It appears at least two different methods are used for moisture analysis, at different temperatures, with different nomenclatures. Why were two different methods used, or two different sets of conditions?
Answer: In fact, the samples were fully dried (105 ºC, 24 h), but for bean analysis it is usual to consider 11% moisture (obtained under 65 ºC, 170h, as a standardized method). We modified the text to become clear.
- Lines 218-221: I think the authors are confusing correlation and causation. Line 221 states that certain macronutrients contribute to genetic diversity. I think they mean to say that genetic diversity is responsible for high variability in those 5 factors, not that the factors themselves contribute to genetic diversity. It is unlikely that they could tie any of these factors as a cause of genetic diversity. I think I understand what they are trying to say, but genetic diversity contributes to variability in these factors, not the other way around.
Answer: The reviewer is correct. We changed the text to become clearer.
- Line 529: The number 13 appears to be bolded and in a different font than the rest of the text.
Answer: It was checked and changed to more adequate form.
Additionally, we revised the formatting of the manuscript and performed minor adjustments some references and other aspects according to the journal requirements.

Round 3
Reviewer 1 Report
The paper has been revised twice, but all the 3 versions have different error bars in the figures. This raises serious ethical concerns about possible data manipulation or fabrication.
Author Response
Comments and Suggestions for Authors
The paper has been revised twice, but all the 3 versions have different error bars in the figures. This raises serious ethical concerns about possible data manipulation or fabrication.
Answer: Dear reviewer. Thanks for raising this issue although your concerns are not justified (and we consider it as offensive to our team), since some of the changes are just a result from the short experience of our Mozambican student and because along the review process we improved the figures (as requested). However, the reviewer is totally mistaken about “ethical concerns about possible data manipulation or fabrication.”. Let’s be very clear on this.
In the first round (the original submitted manuscript) the figures had no error bars, because we feel that it was enough to show only the statistical indexes. After the review it was asked by a reviewer to add the error bars. Therefore in the 2nd version (to which the reviewer 1 is now raising these questions) we introduced the error bars while maintaining as well the statistical indexes. Note that the latter ones did not change. In this way, the reviewer 1 have only 2 versions (not 3) and in the 2nd one we had introduced the error bars as requested. Therefore, the simple suggestion of “ethical concerns about possible data manipulation or fabrication” are misplaced here (and, again, quite offensive).
Although it is not usual for this simple data, we placed in the supplementary material an excel file with the raw data from Figs 2-4, so reviewer 1 have the possibility to check point-by-point the values from those figures if he/she wishes.
We also wish to highlight something quite odd to us. In the previous round reviewer 1 has positively classified our manuscript with “yes” in all categories of the Review Report Form, namely in the questions:
Is the research design appropriate?
Are the methods adequately described?
Are the results clearly presented?
Are the conclusions supported by the results?
The classification of these questions was now changed in round 3 to “Must be Improved”. Why? In fact, we performed all the requested modifications and clarified some points (and even added some new text in the methods as requested by reviewer 2, which accepted these changes and the manuscript). Therefore, at least the classification from reviewer 1 should have been maintained, not to become worst from round 2 to round 3.
It should be emphasized that regarding other potential issues, the manuscript was improved along the review process/rounds and we accepted ALL suggestions pointed by both reviewers (e.g., greater detail in mineral analyse methods, despite the presentation of the support references of these classical methods), and the English language was revised, being quite clear.

Reviewer 2 Report
This paper has come a long way and I appreciate the changes that the authors have made.
I recommend rephrasing the sentence in lines 455-456 for more clarity.
I also recommend that the authors consider more up-to-date methods for future publications (ICP-MS, ICP-OES), as this would greatly raise the relevance and impact of their work.
Author Response
Comments and Suggestions for Authors
This paper has come a long way and I appreciate the changes that the authors have made.
Answer: We deeply thank the reviewer for his/her time spending in improving our manuscript, and for the positive criticisms.
I recommend rephrasing the sentence in lines 455-456 for more clarity.
Answer:
The mentioned sentence in the PDF was:
For S, very strong positive correlations were found for Bean × NAB, Bean × NAF, and Fruit weight × Bean weight; and strong positive correlations for Husk × NAH, Husk × NAF, %Bean × Fruit weight, and %Bean × bean weight.
We altered this sentence and verified (and modified/corrected) all the sentences regarding the correlations of the minerals (Table 7). We hope that this version is clearer
I also recommend that the authors consider more up-to-date methods for future publications (ICP-MS, ICP-OES), as this would greatly raise the relevance and impact of their work.
Answer:
Thanks again for this suggestion. We will surely take it in consideration in future analytical work.
